# Polyphenols and CRISPR as Quorum Quenching Agents in Antibiotic-Resistant Foodborne Human Pathogens (*Salmonella* Typhimurium, *Campylobacter jejuni* and *Escherichia coli* 0157:H7)

**DOI:** 10.3390/foods13040584

**Published:** 2024-02-15

**Authors:** Inocencio Higuera-Ciapara, Marieva Benitez-Vindiola, Luis J. Figueroa-Yañez, Evelin Martínez-Benavidez

**Affiliations:** 1Dirección de Investigación y Desarrollo, Universidad Anáhuac Mayab, Mérida 97310, Yucatán, Mexico; 2Facultad de Ciencias, Universidad Nacional Autónoma de México (UNAM), México City 04510, Mexico; marbenvin@ciencias.unam.mx; 3Centro de Investigación y Asistencia en Tecnología y Diseño del Estado de Jalisco, A.C., Guadalajara 44270, Jalisco, Mexico; lfigueroa@ciatej.mx (L.J.F.-Y.); emartinez@ciatej.mx (E.M.-B.)

**Keywords:** polyphenols, CRISPR, quorum sensing, Quorum Quenching, antibiotic-resistance, foodborne bacterial pathogens

## Abstract

Antibiotic resistance in foodborne pathogens is an increasing threat to global human health. Among the most prevalent antibiotic-resistant bacteria are *Salmonella enterica serovar* Typhimurium, *Campylobacter jejuni* and *E. coli* 0157:H7. Control of these and other pathogens requires innovative approaches, i.e., discovering new molecules that will inactivate them, or render them less virulent without inducing resistance. Recently, several polyphenol molecules have been shown to possess such characteristics. Also, the use of CRISPR (Clustered Regularly Interspaced Short Palindromic Repeats) approaches has recently been proposed for such purpose. This review summarizes the main findings regarding the application of both approaches to control the above-mentioned foodborne pathogens by relying on Quorum Sensing interference (Quorum Quenching) mechanisms and highlights the avenues needed for further research.

## 1. Introduction

The sanitary burden from foodborne diseases globally is a mounting public health issue. Indeed, such an effect is not only limited to decreased productivity and short-term effects but can also lead to a series of chronic diseases as well as mid- and long-term physiological disorders such as Reactive Arthritis, Guillain–Barré Syndrome, Chronic Gastrointestinal Disorders, Kidney Failure, and many others that have an enormous impact on the cost of public health services as well as on lost hours of productive work [1]. Among the pathogenic bacteria that cause most outbreaks globally are *Salmonella enterica serovar* Typhimurium, (*Salmonella* Typhimurium) *Campylobacter jejuni* and *E. coli* 0157:H7. These etiological agents vary in prevalence and impact in different regions of the world, but the statistics available today point to such pathogens as the most impactful ones. In the European Union, for instance, out of 857 notifications registered during 2022 in the Rapid Alert System for Food and Feed (RASFF), 70% were caused by *Salmonella*, 16% by *Listeria monocytogenes* and 5% by *E. coli* [2]; whereas in the US, *Campylobacter* and *Salmonella* are the leading causes of foodborne followed by *E. coli*, *Yersinia*, *Vibrio* and *Cyclospora* [3].

Antibiotic resistance (AR) is a major global public health issue due to a concurrence of causes, among which the following stand out: (1) treatment of animals for disease control, (2) sub-therapeutic use as prophylactic and growth-promoting factors in farmed animals, (3) presence of antibiotic residues in food and pollution of water, marine and terrestrial environments and (4) widespread distribution of genes or plasmids that can be transmitted horizontally and encode for antibiotic resistance. It is well-documented that, as the use of antibiotics intensified and expanded to different ecological niches, microorganisms began to develop resistance mechanisms as part of their survival and evolutionary strategies [4]. Nowadays, this increase in resistance is considered by the World Health Organization (WHO) as a very serious imminent threat to public health globally, estimating that, by the year 2050, there could be up to 10 million deaths attributed to so-called “superbugs” or multidrug-resistant bacteria [5]. It is also worth mentioning that this situation is aggravated by Global Climate Change [6].

Indeed, the emergence of strains resistant to antibiotics is a great cause of public concern as these bacteria can disperse to natural environments such as water and soil, polluting them and causing infections in other animals that can then be transmitted to humans through their products, water, wind, and fecal matter. Antibiotic encapsulation, the modification of the drug’s target gene, dense biofilm formation, and the removal of the antibiotic through an intracellular expulsion pump, are some of the mechanisms for antibiotic resistance; among these, biofilm formation is the most frequent and hardest to overcome as the external polysaccharide matrix significantly limits the antibiotic’s penetration intracellularly [7]. Thus, biofilm formation as a consequence of Quorum Sensing (QS) plays a central role in the development of AR [8,9]. In the food processing industry, biofilm formation and resistance to antimicrobial agents constitute a pressing issue [10,11,12] and requires the development of new strategies and innovative products for sanitization and treatment of surfaces in contact with food during processing.

QS is the mechanism that bacteria use to regulate gene expression in relation to population density through signaling molecules [13,14]. It allows communication among bacteria to coordinate group behavior and is used by pathogens in virulence processes associated with infections and diseases. Group communication and behavioral synchronization allow bacterial populations to develop rapidly, facilitate access to nutrients and ensure higher levels of virulence and improved survival opportunities and fitness. QS involves the activation of specific genes in a coordinated manner in order to synthesize compounds and phenotypic responses to facilitate survival through several metabolic processes, such as biofilm formation, bioluminescence, and increased motility, among others [15].

The standard QS pathway consists of an increase in the bacterial population density, followed by an increase in the concentration of autoinducers (AI) or signaling molecules that are secreted into the environment, including N-Acyl-Homoserine lactones (AHLs) and autoinducers 2 (AI-2) in Protobacteria, as well as short chain peptides in the case of Firmicutes. After reaching a certain concentration threshold and accumulating in a confined environment, the signaling molecules become detectable by bacterial populations, which in turn cause the activation of genes that express proteins involved in several physiological processes such as virulence factors synthesis and horizontal gene transfer. This last mechanism is closely related to the development of resistance to some of the most widely used antibiotics. Furthermore, the communication capabilities offered by QS are very useful because they allow bacteria to acquire traits found in plants, animals, and other higher-level organisms. In short, QS consists of three steps: (1) synthesis and secretion of the signaling molecule and detection of the population density threshold; (2) detection of the signaling molecule by the specific receptor in the bacterial cytoplasm and formation of the receptor-signaling molecule complex; and (3) activation of the specific gene regulating the QS system and triggering the synchronized activity of the bacterial populations [16]. The following elements are involved in this system: (a) autoinducers; (b) signal synthetase; (c) signal receptor; (d) signal response regulator; and (e) genes that regulate the process (QS-Regulon). QS systems used by bacteria can be classified into three groups: (a) Lux/I systems, used primarily by Gram (−) bacteria for AHL synthesis; (b) peptide signaling systems used by Gram (+) bacteria; (c) and the LuxS/AI-2 systems used for communication between different bacterial species [17]. Godinez-Oviedo et al. (2019) analyzed *Salmonella enterica*’s epidemiology, prevalence, and resistance to antimicrobial agents during the 2000–2017 period in México. They report that according to the National Epidemiological Surveillance System (SINAVE), the estimated number of infection cases for *Salmonella typhi* and *Salmonella paratyphi* was 45,280 and 12,458 respectively. The vast majority of Nontyphoidal *Salmonella* (NTS) isolates found could be attributed to the Typhimurium multi-resistant serotype, meaning this pathogen’s impact on Mexican public health is of great relevance, especially when considering that the estimated number of cases adjusted to the sub-reporting increases to over 4.5 million [18,19].

A recently published comprehensive review describes the metabolic pathways specific to each of the microorganisms of importance that are transmitted by zoonosis, recognizing that food is the most frequent transmission route for *Salmonella* Typhimurium and Enterotoxigenic *E. coli* (ETEC) 0157:H7. In this review, the authors detail the QS system, the role it plays on the level of virulence, the underlying genetic mechanism, the role those factors play in the development of antibiotic resistance, and the mechanisms by which this process occurs (biofilm formation, cell expulsion pump, formation of small variants) [20]. Both *E. coli* and *Salmonella* Typhimurium express a receptor responsible for the detection of AHLs, SdiA, that is analog to LuxR; however, while they lack the expression of AHL synthases, both microbes can detect AHLs produced by other pathogenic bacteria like *Yersinia enterocolitica.* It is also known that SdiA regulates an operon located in a virulence gene and the production of biofilms in the presence of AHLs [21,22].

In recent years, the “Quorum Quenching” (QQ) strategy has opened up new possibilities to combat AR [23]. QQ occurs when the QS system is interrupted and the signals switch between microorganisms are inhibited. Competitive inhibition of enzymes and other compounds relevant to QS and interference of signaling molecules throughout the pathway from synthesis, diffusion, grouping to signal detection, are among the most common QQ mechanisms. QQs’ three metabolic pathways are: (1) Production of a signal inhibitory molecule; (2) Degradation of the signaling molecule; (3) Transmission of inhibition by a signaling molecule or (4) Receptor blockage. These mechanisms target microorganisms for destruction or inhibition of the AR development process [24].

Based on the mechanisms described in this review and earlier work, it is possible to achieve QQ using polyphenolic compounds that arrest any of the stages involved at the signaling molecules level, their receptors, and/or the induction of downstream biochemical signals. Also, the use of CRISPR offers an alternative approach that deserves further discussion.

## 2. Polyphenols as Quorum Quenching Agents

Plants are one of the main natural sources of Quorum Quenching Agents (QQA). Alkaloids, phenolic acids, flavonoids, quinones and tannins, terpenes, glucosinolates and lectins are known for their defense properties against herbivores and microorganisms [25]. These compounds have all been shown to decrease or inhibit the virulence of pathogens through various mechanisms that differ from those of antibiotics [26]. Both extracts and molecules of various types of fruits, vegetables, and herbs from different species, have proven to inhibit QS [13,17,27,28]. Several studies have demonstrated that (poly)phenolic compounds act as QS inhibitors [13,29,30]. A review by Nazzaro et al. (2013) presents a list of phytochemicals with proven activity for QS inhibition [17]; later, Nazzaro et al. described their effect on intercellular communication and described the specific (poly)phenols that exert the greatest potential for QS inhibition [31,32]. Molecular studies (PCR, Polymerase Chain Reaction) showed that the diminishing effect on virulence could be attributed to the downregulation of genes involved in QS activation and the development in several bacterial strains. Moreover, the plants’ bioactive compounds can block the synthesis of signaling molecules by AHL-synthase (LuxI), degrading the signaling molecules and/or interacting with the receptor of the LuxR signal (Figure 1). Commonly identified mechanisms can be correlated by their similarity in chemical structure of the QS (AHL) signals well as by their capacity to degrade the signal receptors (LuxR/LasR) [17,33].

Molecules such as catechins have a negative impact in the transcription of various genes related to QS (lasl, lasR, rhll, lasB y rhlA). Indeed, the use of biosensors for RhlR and LasR showed that catechins have an interfering effect on RhlR’s perception of the N-butanoyl-L-homoserine lactone signal, which leads to a reduction in QS factor production. As such, catechins along with other flavonoids produced by higher plants, could constitute a first line of defense against attacks by pathogens through their effects on QS mechanisms and, therefore, on the production of virulence factors [34]. On the other hand, it has also been shown that compounds commonly produced by several plant species that present a gallic acid residue (such as epigallocatechin gallate, ellagic acid, and tannic acid) block AHL-mediated bacterial communication.

An in-depth and updated review on QS interference and elucidated mechanisms and targets of QS inhibitors in AHL-mediated systems is provided in the work by E.M.F. Lima et al. (2023) particularly for *Pseudomonas aeurginosa*, *B. glumae*, *C. volaceum*, *B. cenoepecia* and *Erwinia cartovora*, including the specific QS inhibitors. Also, an overview of recent studies regarding probable modes of action of phenolic compounds on AHL-mediated QS in bacteria is provided [35].

## 3. *Salmonella enterica* Serovar Typhimurium Quorum Sensing Interference with Polyphenols

*Salmonella* Typhimurium is one of the most relevant foodborne etiological agents worldwide. The review carried out by Fan et al. (2022) provides a detailed description of all the mechanisms by which QS is developed in this pathogen, i.e., (1) AHL-mediated QS; (2) Autoinducer-2 (AI-2)-mediated QS; (3) Autoinducer-3 (AI-3)-mediated QS; (4) Indole-mediated QS and 5) Diffusing Signaling Factors (DSF)-mediated QS [20]. As mentioned before, *Salmonella Typhimurium* possesses three types of autoinducers. However, since the Luxl gene is lacking, it is through a homolog of LuxR (SdiA) that the detection of signaling molecules occurs [36,37]. The rck operon is regulated by the SdiA protein in the presence of AHL, and thus this operon plays an important role in cell entry [38]. AI-2 regulates the expression of genes related to antioxidative bacterial stress (soda, sodCl and sodCII), flagella (fliC and fliD) and the molecule Type 3 Secretion System-1 (T3SS1) located in *Salmonella* Pathogenicity Island-1 (SPI-1), which is involved in bacterial invasion and survival. A1-3 regulates the expression of T3SS1 and T3SS2, whose activity influences bacterial invasion capabilities and intracellular survival, respectively. AI-3 can also regulate biofilm formation through regulation of flagellar development. AHLs regulate the expression of srgE and rck loci, which in turn affect fimbria, bacterial invasion and complement resistance mechanisms. Indole mediation represses SPI-1 expression and DSF thus inhibiting SPI-1 expression. Such characteristics also relate to virulence capability. On the other hand, regarding antimicrobial agents’ resistance, AHL and AI-3 indirectly regulate biofilm formation through their effects on flagella and the rck operon in the pRDT98 plasmid, respectively. AHLs also mediate AR by affecting membrane protein expression. Indole can affect the expression of the AcrB efflux pump and mediate AR through the induction of oxidative stress pathways and phage shock response. Additionally, DSF- and indole-mediated QS have also been reported in *S. enterica serovar* Typhimurium and are known to play a fundamental role in virulence regulation and AR [10,14,38,39,40,41,42].

*Salmonella* Typhimurium is frequently transmitted through food or water and is also considered relevant to resistance transmission [43]. The studies carried out by Zaidi et al. (2007) have documented the severity of multi-resistance in this pathogen and its rapid advance since 2007, as well as contamination of some highly consumed foods [44,45]. Also, Cloeckaert and Schwarz (2001) detailed the multi-resistance patterns of *Salmonella Typhimurium* DT (Definitive type) 104 underscoring the importance of genes the role of genes encoded in the chromosome and conferring resistance to ampicillin, chloramphenicol, streptomycin, sulfonamides and tetracyclines [46].

Several polyphenols have been identified as growth inhibitors of *Salmonella Typhimurium*, including coumarin [47], carvacrol, trans-cinnamaldehyde, β-resorcylic acid, and eugenol [48]. In 2013, using docking scores, Gnanendra et al. (2013) identified three compounds with outstanding potential to interfere with QS in *Salmonella* Typhimurium [49]. Also, Alvarado-Martínez et al. (2020) have shown that phenolic acids such as gallic acid, protocatechuic acid and vanillic acid inhibit *Salmonella* Typhimurium growth through alteration of virulence genes and increased membrane permeabilization [50]. Intracellular survival of *Salmonella* Typhimurium was found to be significantly affected by pyrogallol, alone or in combination with marbofloxacin. In their study, the authors showed downregulation of several virulence genes and inhibition of the expression of the SdiA and rck genes [51]. The role of thymol, the most abundant polyphenol in oregano, was shown to possess inhibitory characteristics against *Salmonella* Typhimurium by reducing biofilm formation ability, disrupting the cell membrane, and downregulating virulence genes [52], while Zhang et al. (2022) showed that virulence factors for Lon protease degradation were also targeted by thymol [53]. Thymol, together with piperine have been shown to possess synergistic activity with kanamycin and streptomycin against *Salmonella* Typhimurium [54]. A recent review of phytochemicals, including some polyphenols, has been published by Almuzaini (2023) and includes synergism among compounds, plant origin, solvent of extraction (methanolic, ethanolic, hydroethanolic and hexanic), type of biological activity exerted as well as the Minimum Inhibitory Concentrations [55].

A very in-depth and recent review by Sakarikou et al. (2020) has summarized all the previous works regarding the specific antibiofilm properties of a wide array of plant extracts, essential oils and chemical compounds including polyphenols, essential oils, plan extracts, and many other phytochemicals targeted against *Salmonella*. Table 1 provides a summary detailing the specific phytochemical, target microorganism, antibiofilm effect as well as mode of the action mechanism. From this summary, the following compounds offer the best polyphenol alternatives for biofilm inhibition by *Salmonella* Typhimurium: carvacrol, dihydroxibergamottin, bergamottin, thymol, eugenol, and gallic acid. Also, an extensive variety of essential oils and plant extracts are provided [56].

## 4. CRISPR and Quorum Quenching in *S. Typhimurium*

The knowledge of QS in *S. typhimurium* allows for the generation of strategies for regulating or controlling QQ in order to develop future technologies for diagnosis, experimentation, and therapy. In the case of *S. typhimurium*, it is not only possible to use the knowledge of the mechanisms involved in the cell signaling process, but also of the genes that actively participate in their response systems and that are part of the molecular recognition and response. Through CRISPR Cas systems, it is possible to edit and regulate mechanisms related to QS, as well as key genes involved in molecular signaling. Some published articles report key factors of QS signaling where tools such as CRISPR Cas-9, dCas9, 12a, 13, 13a, and 14 a1 can be used to regulate the virulent effects of bacteria. Kiga et al. 2020, demonstrated the potential and effectiveness of the use of CRISPR-type techniques to promote useful strategies for the regulation of signaling in pathogenic microorganisms such as *E. coli* and *S. aureus* and achieve effective QQ [57,58], while the studies of Sharma et al. (2022) carried out on *S.* Typhimurium knockout lines inactivated through a single step and using PCR products, according to the method of Datsenko and Wanner (2000), who demonstrated by Reverse-Transcription Polymerase Chain Reaction (RT-PCR) the repression of the genes that code for flagellum (fliC, flgK) and fiber (csgA) due to the activation of the endogenous CRISPR-Cas system that regulates the differential formation of films and biofilms that adhere to the surface [59,60]. So far, this work seems to be the only one using this approach for the specific application to control *Salmonella* Typhimurium.

Ma et al. 2023 developed an innovative fluorescent radiometric biosensing platform, called SCENT-Cas (Silver nanoCluster Empowered Nucleic acids Test using CRISPR/Cas12a) specially designed for *S. typhi* capable of sensitively and specifically detecting pathogenic bacteria in complex samples. SCENT-Cas took advantage of the qualities of the CRISPR/Cas12a system; at the same time, it converted target nucleic acid signals into two-color fluorescence using label-free DNA-templated AgNCs. The time between sampling and response was approximately 2.0 h, which allowed for the rapid decision making and response to treatment of the pathogenic bacterial infection. In general, the method consisted of isothermal amplification of the InvA gene using LAMP, which is specific to the *S. typhi* species, which subsequently triggered CRISPR/Cas12a trans cleavage. Knowing in depth the function of the genes, operons or other important elements involved in QS provides the necessary tools to develop diagnostic devices for the detection of important groups of pathogenic bacteria. This process mentioned above can lead the way towards the discovery of new therapeutic targets combined with the effectiveness of CRISPR systems [61]. Wang et al. 2023 published a very complete review, which mentions next-generation diagnostic tools for the detection of diseases in humans, caused by different factors, including the bacteria mentioned here. The editing tools used so far, as well as the detection of proteins and other molecules, are mentioned in detail [62].

## 5. Polyphenols as QQ Agents in *E. coli*

*Escherichia coli* is a Gram-negative bacterium that is facultative anaerobic in nature and is known to form biofilm on various surfaces. Biofilms formed by toxigenic *E. coli* O157:H7 are typically surface-attached arrangements of cells that are embedded within the self-produced matrix of extracellular polymeric substance (EPS) [63]. The biofilm formation by *E. coli* contributes to the occurrence of various infections and makes their eradication difficult [64].

Several investigations report the inhibition of *E. coli* O157:H7 biofilms by various extracts of plants and single plant compounds. Single phenolic compounds known to inhibit biofilm formation in *E. coli* in vitro include 4-hydroxybenzoic, syringic, gallic, vanillic, cinnamic, and p-coumaric acids, (+)-catechin, (−)-epicatechin, quercetin, polydatin, and resveratrol [65]. Other phytochemicals, such as the flavonoid phloretin, a major compound in apple and strawberry extracts [66], and two furocoumarins (bergamottin and dihydroxybergamottin) isolated from grapefruit juice [67] as well as trans-resveratrol from red grapes and grapefruit seed extracts, have also been shown to inhibit the formation of *E. coli* O157:H7 biofilms [68,69].

Biofilm formation in *E. coli* is a highly regulated process controlled by several factors, including autoinducer-mediated cell–cell signaling. Differential responses for different flavonoids were observed for different cell–cell signaling systems. Among the tested flavonoids, naringenin, kaempferol, quercetin and apigenin were effective QS antagonists and biofilm suppressors in the *E. coli* O157:H7 strain [27]. For non-O157 Shiga toxin producing *E. coli* strains, Sheng et al. (2016) found that the grape seed extract inhibited the QS system well [70].

During the early phase of biofilm development, the adhesive organelles, such as type I fimbriae (pili) and curli fimbriae, play a major role in the irreversible attachment of *E. coli* to the surface [71]. Curli are adhesive amyloid fibers present on the cell surface of *E. coli* that help to maintain cell–cell and cell–surface interactions and lead to biofilm formation [64,72,73,74]. Pili are extracellular adhesive fibers, which mediate biofilm formation, binding, and invasion into host cells. Some studies have evaluated the ability of phytochemicals to inhibit curli and pili to prevent the formation of *E. coli* biofilm [62]. Ginkgolic acid, vitisin B, coumarin, umbelliferon, and eugenol have shown significant inhibition of *E. coli* biofilm formation by the downregulation of curli genes or genes that contribute to formation of biofilm [75,76,77,78]. *E. coli* has flagellae that contribute to motility depending on the environment and may be an essential part of inducing microbial adhesion on the host surface, allowing biofilm formation [79,80]. The blueberry extract was outstanding in the inhibition of the QS inhibition related to swarming motility in *P. aeruginosa* and *E. coli* O157:H7 pathogens [81].

Some compounds like condensed tannins (procyanidins and prodelphinidins) showed antimicrobial activities against *E. coli* serotype 078 by affecting the growth biofilm formation and motility [80] and phenolic acids (gallic acid and ferulic acid) inhibited bacterial motility of *E. coli* CECT 434. Both gallic acid and ferulic acid caused total inhibition of swarming in *E. coli* and thus reduced the biofilm mass considerably [82]. Other studies show that the β-sitosterol glucoside isolated from citrus fruit inhibits the biofilm formation and motility through rssAB- and hns-mediated repression of flagellar master operon flhDC in *E. coli* O157:H7 [83]. ε-viniferin, a derivative of resveratrol, showed the downregulation of important genes such as flhD, fimA, fimH, and motB, which are involved in motility regulation and adherence of cells to surfaces [68,84]. The antimicrobial strategies based on the inhibition of QS represent a key tool for the control of antibiotic resistance and to inhibit virulence factor expression. Bai et al. (2022) evaluated the ability of echinatin and gingketin to inhibit QS, formation of biofilms, motility, and synthesis of virulence factors. Also, they assessed the synergistic effect with the colistin B and colistin E and gentamycin antibiotics, proving a significantly greater antimicrobial activity against *E. coli* O157:H7 and five other clinical isolates (*E. coli* C 83654, *E. coli* XJ 24, *E. coli* O101, *E. coli* O149, *E. coli* KD-13-1) [85,86]. Also, Ivanov et al. (2022) evaluated the effect of 11 polyphenols on different bacterial strains, including *E. coli* IBRS E003 and *E. coli* IMD989. Their results showed that polyphenols significantly reduced planktonic-cell and biofilm growth in antibiotic resistant strains [87]. The general information concerning inhibitory action of polyphenols against *E. coli* is provided in Table 2.

## 6. CRISPR and QQ in *E. coli* 0157:H7

As mentioned above, knowing the functional elements of the QS is of great importance, such as the sdiA gene, which controls the virulence factors EspD and intimin in *E. coli* O157:H7 [88]. With the emerging technology of CRISPR Cas, the possibility of improving systems and devices for detecting microorganisms of interest in the food, medical and environmental industries becomes evident. Zhu L. et al., 2023, published the development of an ultrasensitive method for the detection of *E. coli* O157:H7 based on what they call RAA-CRISPR/12a (Recombinase-Aided Amplification, RAA), resulting in a highly efficient test that only requires 55 min to detect *E. coli* O157:H7 [89]. Recently, Jiang et al., 2023, developed a rapid, specific, and visual nucleic acid detection method called CRISPR/Cas12a-PMNT in *Escherichia coli* O157:H7, based on a combination of Cas12a techniques with RPA (Recombinase Polymerase Amplification) and cationic poly water-soluble [3-(3′-N,N,N-triethylamino-1-propyloxy)-4-methyl-2,5-thiophene hydrochloride] (PMNT). They finally reported that the Cas12aVIP method produced high specificity and did not interfere with other non-target bacteria. This method ensures that the detection was performed within 40 min and that the signal can be observed with the naked eye under natural light, which presents great potential for multiple applications of rapid detection of nucleic acids without the need for technical expertise or auxiliary equipment [90].

Zhang R., et al. (2021) described the advantages and disadvantages that the CRISPR interference system has for the silencing or regulation of multiple or specific genes in *E. coli*; in this study, the authors emphasize the importance of developing these type of tools for the understanding of pathogenesis and its future treatment [91].

In a letter to the editor, Suvvari et al., 2023 mention that there are some research groups that use the CRISPR-Cas13a system to attack antimicrobial resistance and thus generate hope for the future. Other work carried out on the topic of gene regulation and silencing is also mentioned, using specific target RNA sequences in *E. coli* and *Leptotrichia Shahii*. It is concluded that this system has enormous potential and still needs to be perfected for use in research, diagnosis, and therapy [92].

## 7. Polyphenols as QQ Agents in *Campylobacter jejuni*

*Campylobacter* is recognized as a major bacterial agent causing gastroenteritis and medium-term effects like reactive arthritis, meningitis, pancreatitis, and Guillain–Barre’s syndrome, on human health worldwide and more particularly, in developed countries [93,94]. *Campylobacter* spp. can be found in water reservoirs, as commensals in the intestinal tract of animals, particularly birds, and as virulent pathogens in humans. Contaminated animal food products, particularly poultry, are a major source of bacteria that cause human campylobacteriosis [95]. Several studies have shown the bacterium’s ability to adhere to inert surfaces of different materials used in different industries [96]. *Campylobacter jejuni*, and *Campylobacter coli*, can form mono- and multi-species biofilms [97]. Plant materials represent an important source of phytochemicals that prevent adhesion and biofilm formation for which bacterial adhesion is the first step. Wagle et al., 2021 reported the effects of turmeric, curcumin, allyl sulfide, garlic oil, and ginger oil on *C. jejuni*. The selected phytochemicals (except curcumin) reduced its adhesion to chicken embryo cells, and all the phytochemicals reduced QS [98]. Other compounds with anti-adhesion and anti-biofilm activities against *C. jejuni* reported in the literature are carvacrol, trans-cinnamaldehyde, epigallocatechin gallate, amentoflavone, β-resorcylic acid, eugenol, linalool, and resveratrol [99,100,101,102,103,104,105,106,107].

Extracts such as blackberry and blueberry pomace significantly reduced the growth of *C*. *jejuni* and altered the physicochemical properties such as cell surface hydrophobicity and auto-aggregation of this bacterial pathogen. *Alpinia katsumadai* extracts, grape extract, thyme extract, and herbal extracts also possess anti-adhesive and anti-biofilm activities [107,108,109,110]. On the other hand, the ability of essential oils (lavender, juniper, rosemary, juniper, clove, thyme, coriander) to inhibit the adhesion and formation has also been reported [105,111].

In *C. jejuni,* cell density phenotypes such as is motility, host colonization, virulence, and biofilm formation are associated with the AI-2-mediated quorum sensing system [112]. Citrus extracts reduce motility, biofilm formation, invasion, and adhesion of epithelial cells and virulence of *C. jejuni* by modulation of QS. Castillo et al. reported that the extract of *Citrus limon*, *Citrus medica* and *Citrus aurantium* peels decreased AI-2 activity and reduced expression of flaA-B and genes involved in adherence and invasion processes (cadF and ciaB) [113,114]. A *Euodia ruticarpa* extract has also shown anti-QS activity, although a link between the reduction of biofilm formation and QS activity was not shown [115]. In their review, Elgamundi and Korolik (2021) also report biofilm inhibition with secoiridoid and hydroxycinnamic and gallic acid as well as with taxifolin, epigallocatechin gallate, and resveratrol [116]. The inhibitory action of polyphenols against *C. jejuni* is provided in Table 3.

## 8. CRISPR and QQ in *Campylobacter jejuni*

Campylobacter, like many other bacteria, contain their own CRISPR Cas systems, which means that their endogenous CRISPR Cas9 systems are not necessarily affected by the delivery of gRNAs towards their endogenous QS elements to be regulated [117]. Abavisami et al., 2023 reported a pair of studies where the enzymes AacCas12b and Cas12a are used to develop devices for the detection of *C. jejuni* through different techniques. Not all the QS signaling molecules involved in *C. jejuni* have been determined; however, it is possible to design dCas9-or dCas13-type tools to attack and regulate the target genes responsible for its virulence such as cadF, ciaB, cdtB and flaA, (Table 4) [118]. Like *S. typhi, C. jejuni* expresses genes from the LuxS family [119,120]; therefore, it is also considered a good target to develop CRISPR-type regulation tools.

Costigan et al. (2022) were able to validate the repression of the astA gen which encodes for arilsulfatase using the CRISPR interference (CRISPRi) in *Campylobacter jejuni* [122]. The same tool was applied to a deleted M1Cam strain showing that the Cas9 endogenous system did not affect the CRISPRi system. Also, a reduction in motility was achieved by affecting the hipO gen and inhibiting the flagellar genes flgR, flaA, flaB. These flagellar variants were confirmed by phenotyping and Electron Transmission Microscopy (ETM). All these techniques are focused toward the diminishment of horizontal and vertical transfer of antibiotic resistance.

In the review carried out by Rodrigues et al. 2023, in the section corresponding to “Campylobacter”, it is mentioned through the compilation of information from various authors, that if the endogenous Cas9 of *C. jejuni* is eliminated, it reduces adhesion, invasion and in vitro bacterial translocation across monolayers in human colorectal adenocarcinoma cells. Cas9 from *C. jejuni* is toxic to human cells, so the *C. jejuni* Cas9 gene, transcript and protein are a perfect target to control or regulate the infectious processes caused by this bacterium. For *Campylobacter*, as well as for other bacteria, once confirmed that it carries a CRISPR-Cas9 type system, it is recommended that, in addition to using other combined CRISPR systems of the type Cas3, dCas9, Cas13a, dCas13 or others, all this is used for the system to be effective (Figure 2) [123].

## 9. Conclusions

Polyphenols are an excellent alternative to combat three of the most dangerous foodborne pathogens analyzed in this review. Many extracts and pure compounds have been characterized and their inhibitory activity against drug resistance has been identified. Most inhibitory mechanisms have to do with the downregulation of virulent genes, inactivation of the SidA protein, interference with mechanisms leading to biofilm formation, and interference with QS signaling molecules. While knowledge about the specific mechanisms leading to such action has been detailed in some reports, much work remains to be done in this respect so that Quorum Quenching becomes a practical tool against antibiotic resistance pathogens. All reviewed results suggest that polyphenols should be considered an important source of alternative antimicrobial control strategies to those relying on antibiotics. On the other hand, the three pathogens contain their own endogenous CRISPR Cas-type systems; therefore, the strategies for their study through non-endogenous CRISPR systems are very variable and complex, since it is necessary to skip the typical endogenous responses that render exogenous CRISPR modification or regulation tools ineffective. It is important to emphasize that the idea of using CRISPR systems in these microorganisms is to be able to understand the various functions of their genetic elements, in terms of application for further research. Another branch of great importance and urgency is the field of diagnosis. In this field, it is necessary to use CRISPR Cas systems different from the commonly known endogenous systems such as dCas9, dCas12, dCas13, and Cas14. Knowing and then attacking the signaling systems of Gram-positive and Gram-negative microorganisms is of vital importance, since the tools that must be directed at these targets must be extremely precise. In 2017, Zuberi et al. accomplished biofilm inhibition using a QS mechanism in *E. coli* using the CRISPRi system [124]. This was achieved by targeting the LuxS gene which translates the synthase involved in the AI-2 inducer. The utilization of CRISPRi systems is efficient because it does not compromise the activation of the endogenous CRISPR systems. Recently, Alshammari et al. (2023) showed that by knocking out the genes involved in QS such as luxS, firmH and boIA, an effective reduction in the synthesis of EPS can be achieved [122,125]. Thus, they confirm that the CRISPR Cas9-HDR (Direct Homologous Repair) tool can be utilized to substitute the genes involved in the QS process through the utilization of the direct homologous repair (HDR).

## Figures and Tables

**Figure 1 foods-13-00584-f001:**
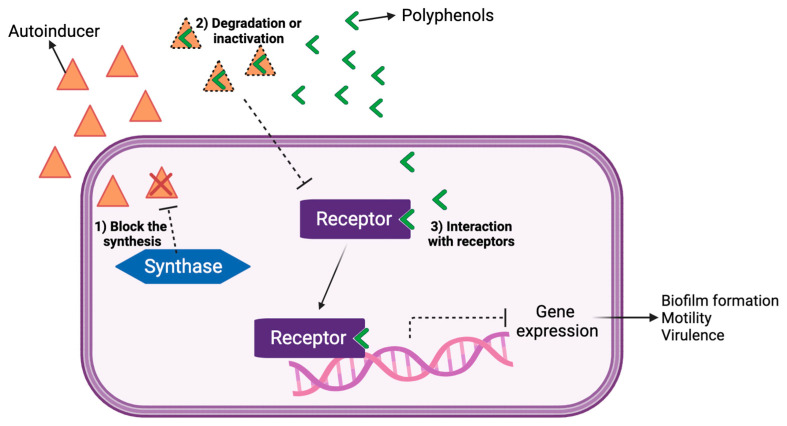
Polyphenol inhibition mechanisms as Quorum Quenching agents. Figure created by BioRender.

**Figure 2 foods-13-00584-f002:**
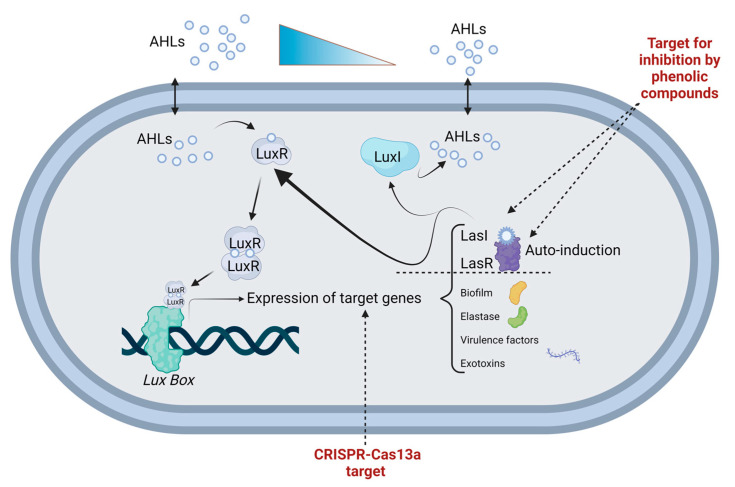
General CRISPR interference mechanism for QS inhibition. Proposed sites to intervene in the QS signaling process with CRISPR-Cas13a and phenolic compounds. Figure created by BioRender.

**Table 1 foods-13-00584-t001:** Inhibitory action of polyphenols against *S. Typhimurium*.

Compound or Extract	Inhibitory Action	Reference
Coumarin	Not specified (Moderate)	[47]
Carvacrol Trans-cinnamaldehyde B-resorcylic acid Eugenol	Increased susceptibility to antibiotics	[48]
CD01374 RJF 004047 KM101117	Inhibition of SDiA activity	[49]
Gallic acid Protocatechuic acid Vanillic Acid	Alteration of virulence genes and increased membrane permeability	[50]
Pyrogallol	Downregulation of virulence genes	[51]
Thymol	Biofilm inhibition, disruption of cell membrane and downregulation of virulence genes	[52]
Thymol	Lon protein degradation	[53]
Thymol and Pipperine	Synergistic effect with kanamycin and streptomycin	[54]

**Table 2 foods-13-00584-t002:** Inhibitory action of polyphenols against *E. coli*.

Compound or Extract	Inhibitory Action	Reference
4-hydroxybenzoic, syringic, gallic, vanillic, cinnamic and p-coumaric acids, (+)-catechin, (−)-epicatechin, quercetin, polydatin and resveratrol. Bergamottin and dihydroxybergamottin Grapefruit seed extract	Inhibited the biofilm formation	[65,67,69]
Phloretin	Inhibited the biofilm formation by impaired autoinducer II expression and fimbriae expression.	[66]
Trans-resveratrol Naringenin, kaempferol, quercetin and apigenin.	Inhibited the biofilm formation by interfere of AI-2 signaling.	[27,68]
Grape seed extract	Suppresses QS with concomitant decrease in motility, flagella protein expression and Shiga toxin production.	[70]
Ginkgolic acid	Repressed curli genes and prophage genes and influenced swarming and swimming motilities.	[75]
Vitisin B	Biofilm inhibition by fimbriae reduction.	[76]
Coumarin and umbelliferon	Repressed curli genes and motility genes.	[77]
Eugenol	Down-regulation of curly (csgABDFG) and type I fimbriae genes (fimCDH) and ler-controlled toxin genes (espD, escJ, escR, and tir), which are required for biofilm formation and the attachment and effacement phenotype.	[78]
Procyanidins and prodelphinidins	Showed antimicrobial activities against *E. coli* by affecting the growth biofilm formation and motility.	[80]
Gallic acid and ferulic acid	Inhibition of swarming in *E. coli* CECT 434 and thus reduced the biofilm mass considerably.	[82]
β-sitosterol glucoside	Inhibit the biofilm and motility through rssAB- and hns-mediated repression of flagellar master operon flhDC in *E. coli* O157:H7.	[83]
ε-viniferin	Downregulation of genes such as flhD, fimA, fimH and motB which are involved in motility regulation and adherence.	[68,82]
Hesperetin, hesperidin, naringenin, naringin, taxifolin, morin, chlorogenic acid, ferulic acid, p-coumaric acid, and gallic acid	Inhibit bacterial growth and biofilm formation of *E. coli* IBRS E003 and *E. coli* IMD989	[87]

**Table 3 foods-13-00584-t003:** Inhibitory action of polyphenols against *C. jejuni*.

Compound or Extract	Inhibitory Action	Reference
Turmeric, curcumin, allyl sulfide, garlic oil, and ginger	Reduced the adhesion (except curcumin) and all the phytochemicals reduced quorum sensing.	[98]
Carvacrol	Inhibited the biofilm formation and adhesion of bacteria and decreasing motility, quorum sensing, and tolerance to stress in vitro. Downregulated bacterial cell mobility genes flaA, flaB, and flaG	[99]
Trans-cinnamaldehyde	Inhibited the biofilm formation and adhesion of bacteria. Downregulated bacterial cell mobility genes flaA, flaB, and flaG.	[99]
Epigallocatechin gallate	Disturbed quorum-sensing activity and reduced motility, adhesion, and biofilm formation.	[101,102]
Amentoflavone	Inhibited of adhesion.	[103]
β-resorcylic acid	Down-regulated expression of genes for motility (motA, motB) and attachment (cadF, ciaB)	[104]
Eugenol	Downregulated of genes (flaA, flaaG, flgA, waaF, cosR, and ahpC) critical for biofilm formation.	[99]
Linalool	Inhibited of adhesion and QS.	[105]
Resveratrol	Inhibited the biofilm formation and adhesion.	[106,107]
Alpinia katsumadai extracts Grape extract Thyme extract Herbal extracts Lavender, juniper, rosemary, cloves, thyme (essential oil)	Inhibited adhesion.	[107,108,109,110,111]
Coriander	Inhibited of adhesion and QS.	[105]
Citrus extracts	Reduce motility, biofilm formation, invasion, and adhesion of epithelial cells and virulence of C. jejuni by modulation of QS.	[113,114]
*Euodia ruticarpa*	Inhibited of QS.	[115]

**Table 4 foods-13-00584-t004:** Effectiveness of CRISPR tools derived from QS and QQ in selected foodborne bacteria (*Salmonella* Typhimurium, *E. coli* 0157:H7 and *Campylobacter jejuni*).

Cas system	Organism	Application	Reference
CRISPR-Cas13	*Salmonella* Typhimurium	Reduction of *S.* Typhimurium colonization in the intestinal tract.	[121]
RAA-CRISPR/12a	*Escherichia coli* O157:H7	Highly efficient diagnostic test development.	[122]
CRISPR-dCas9	*Campylobacter jejuni*	Verified the variation in the reduction of mobility that exists in flagellum phenotypes	[123]

## Data Availability

The data used to support the findings of this study can be made available by the corresponding author upon request.

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
