# Peer review of "Polyphenols and CRISPR as Quorum Quenching Agents in Antibiotic-Resistant Foodborne Human Pathogens (Salmonella Typhimurium, Campylobacter jejuni and Escherichia coli 0157:H7)"

_foods, 2024, doi:10.3390/foods13040584_

Round 1

Reviewer 1 Report

Comments and Suggestions for Authors

Dear Authors,

The review manuscript entitled “Polyphenols and CRISPR as Quorum Quenching Agents in Antibiotic-Resistant Foodborne Human Pathogens (Salmonella Typhimurium, Campylobacter jejuni and Escherichia coli 0157:H7).” is appropriately well-written and developed by Higuera-Ciapara et al. in suitable English with a clear structure. They reviewed recent advances and studies in applications of different Quorum Sensing mechanisms against some drug-resistant foodborne pathogens. This review paper is novel and interesting. Some points of view should be clarified and revisions need to be addressed by the authors as described below:

-        Using graphical abstracts is very common and practical in professional review papers. The authors should present a high-quality and comprehensive professional graphical abstract including the mechanisms against foodborne pathogens discussed in this study in summary.

-        Please merge the discussion section into the other previous sections in the manuscript.

-        Add a section regarding the applications of these novel antibacterial mechanisms and the potential of using these methods in food processing and food products as practical methods to provide food safety.

-        Remove names of different foodborne pathogens in the keywords section. Only use “Foodborne bacterial pathogens”. 

Author Response

Dear Sirs,
I hereby enclose the revised manuscript for the review titled "Polyphenols and CRISPR as Quorum Quenching Agents in Antibiotic-Resistant Foodborne Human Pathogens (Salmonella Typhimurium, Campylobacter jejuni and Escherichia coli 0157:H7)". we kindly hope you will take it into consideration.

Best regards,

Reviewer 2 Report

Comments and Suggestions for Authors

The article titled "Antibiotic Resistance in Foodborne Pathogens: Approaches to Control Using Quorum Sensing Interference" discusses a critical issue in public health, which is antibiotic resistance in foodborne pathogens. The article highlights that antibiotic resistance poses a growing global threat to human health, and it specifically focuses on three prevalent antibiotic-resistant bacteria: Salmonella enterica serovar Typhimurium, Campylobacter jejuni, and E. coli 0157:H7. The paper suggests that innovative approaches are required to combat these pathogens, such as the discovery of new molecules that can inactivate them or reduce their virulence without inducing resistance.

The article also mentions the recent exploration of polyphenol molecules and CRISPR approaches as potential solutions to control these foodborne pathogens. It underscores the application of Quorum Sensing interference (Quorum Quenching) mechanisms in addressing antibiotic resistance and highlights the need for further research in this area.

Why is antibiotic resistance in foodborne pathogens considered an increasing threat to human health globally, and what are the implications of this threat?

Why are Salmonella enterica serovar Typhimurium, Campylobacter jejuni, and E. coli 0157:H7 highlighted as particularly prevalent antibiotic-resistant bacteria in this context?

Why is it crucial to explore alternative approaches, such as the use of polyphenol molecules and CRISPR techniques, in controlling these pathogens?

Why is Quorum Sensing interference (Quorum Quenching) mentioned as a potential mechanism for combating antibiotic resistance, and how does it work?

Why is further research needed in the field of antibiotic resistance in foodborne pathogens, and what are the specific research avenues that the article suggests for future investigation?

Comments on the Quality of English Language

The quality of English in the provided article is generally good

Author Response

(The authors gave the same response as above.)

Reviewer 3 Report

Comments and Suggestions for Authors

The paper is dedicated to a critical issue nowadays - the issue of antibiotic resistance which is increasing annually. This paper will widen the knowledge of readers. Moreover, this paper is closely related to herbal active substances which could be a good alternative to antibiotics or additives to antibiotics

I have some recommendations for improving this manuscript

1. Line 17 and 123 CRISPR - please, explain this abbreviation

2. Lines 135,138,140,143 - explain abbreviation 

3. Lines 161 and 166 - abbreviation should be decoded in line 161

4. Lines 161 and 168 - the same 

5. Line 205 - it is better to say solvent of extraction. Type means the type of extraction leki this maceration, remaceration, etc.

6. Line 204 delete the part in order to read easier

7. Line 208-209 it is necessary to paraphrase or modify the sentence paying attention that extracts and essential oils are complex mixtures of substances

8. 210-211 - it is not clear what is written

9, Table 2 the last line -what is this

10. There are no conclusions.

Probably, the part of the section discussion could be conclusions

Dear authors, please check the whole text concerning abbreviations and their decoding to improve the readability of the manuscript. Secondly, it is necessary to write introductory sentences to tables like this. In addition, the table of abbreviations could be good

The generalization information concerning..... is provided in table ....

Comments on the Quality of English Language

The language is good. The manuscript is easily readable. However, there are a lot of abbreviations. I wish the authors would make the table of abbreviations

Author Response

(The authors gave the same response as above.)

Round 2

Reviewer 1 Report

Comments and Suggestions for Authors

Dear authors,

Thank you for your responses and revising.

I have no more comments.